# Quantification of the capacity of *vibrio fischeri* to establish symbiosis with *Euprymna scolopes*

**Aidan R. Donnelly**[1☺]**, Elizabeth J. Giacobe**[1☺]**, Rachel A. Cook**[1]**, Gareth M. Francis**[1]**, Grace K. Buddle**[1]**, Christina L. Beaubrun**[1]**, Andrew G. Cecere**[1]**, Tim I. Miyashiro**[1,2]*

**1** Department of Biochemistry and Molecular Biology, The Pennsylvania State University, University Park, PA, United States of America, **2** The One Health Microbiome Center, Huck Institutes of the Life Sciences, The Pennsylvania State University, University Park, PA, United States of America

☺ These authors contributed equally to this work.

\* tim14@psu.edu

**Data Availability Statement:** All relevant data are within the paper and its Supporting Information files.

## Abstract

Most animals establish long-term symbiotic associations with bacteria that are critical for normal host physiology. The symbiosis that forms between the Hawaiian squid *Euprymna scolopes* and the bioluminescent bacterium *Vibrio fischeri* serves as an important model system for investigating the molecular mechanisms that promote animal-bacterial symbioses. *E. scolopes* hatch from their eggs uncolonized, which has led to the development of squid-colonization assays that are based on introducing culture-grown *V. fischeri* cells to freshly hatched juvenile squid. Recent studies have revealed that strains often exhibit large differences in how they establish symbiosis. Therefore, we sought to develop a simplified and reproducible protocol that permits researchers to determine appropriate inoculum levels and provides a platform to standardize the assay across different laboratories. In our protocol, we adapt a method commonly used for evaluating the infectivity of pathogens to quantify the symbiotic capacity of *V. fischeri* strains. The resulting metric, the symbiotic dose-50 ($SD_{50}$), estimates the inoculum level that is necessary for a specific *V. fischeri* strain to establish a light-emitting symbiosis. Relative to other protocols, our method requires 2–5-fold fewer animals. Furthermore, the power analysis presented here suggests that the protocol can detect up to a 3-fold change in the $SD_{50}$ between different strains.

## Introduction

Animal-bacterial symbioses are ubiquitous in nature, with many bacterial symbionts contributing to the physiology, development, and even behavior of the animal host [1–3]. The mutualistic symbiosis established between the Hawaiian bobtail squid, *Euprymna scolopes*, and the marine bacterium, *Vibrio fischeri*, has served as an important model for the study of animal-bacterial symbioses [4]. Populations of *V. fischeri* are housed within a symbiotic organ (light organ), where they receive host-derived nutrients and energy sources in exchange for producing bioluminescence that permits the host to engage in an anti-predatory behavior called counterillumination [5–8]. The symbiosis is initially established in juvenile squid, which hatch with their light organs uncolonized (apo-symbiotic) and acquire *V. fischeri* symbionts from

**Funding:** This work was supported by the National Institute of General Medical Sciences Grant R01 GM129133 (to T.I.M.). The funder did not and will not have a role in study design, data collection and analysis, decision to publish, or preparation of the manuscript.

**Competing interests:** The authors have declared that no competing interests exist.

the marine reservoir [9]. The abundance of *V. fischeri* has been reported to range from ~1 CFU/mL (seawater) to >100 CFU/mL (sediment) [10], although the distribution of individual strains remains unclear. Within as little as 8 hours after being exposed to *V. fischeri*, *E. scolopes* juveniles will begin emitting bioluminescence [11]. The ability to culture and genetically modify *V. fischeri* [12] and the ease with which *E. scolopes* juveniles can be generated in a lab setting [13] have led to the development of simple squid-colonization assays that have elucidated the molecular underpinnings of symbiosis establishment [14].

Previous studies have shown that the inoculum conditions, *e.g.*, strain type, concentration of cells within the inoculum, and its duration, have a significant impact on whether hatchlings establish symbiosis [11, 15]. Therefore, knowledge of how the inoculum concentration affects the ability of a strain to establish symbiosis has the potential to reveal biological insight and inform subsequent studies. For instance, if one strain can establish symbiosis using an inoculum at a lower abundance than another strain, then performing the colonization assay with each strain at a low abundance may lead to the false conclusion that only one strain is capable of symbiosis. Consequently, efforts to develop simple assays to evaluate the extent to which a strain can establish symbiosis will provide the ability to justify the inoculum levels used in subsequent studies. Furthermore, because research involving cephalopods frequently falls under the auspices of institutional animal care and use committees (IACUC), these efforts can also help reduce the number of animals used in a project, which is one of the 3Rs principles for conducting ethical and humane scientific studies involving animals [16, 17].

The goal of the protocol described here is to report a metric that quantifies the ability of a strain of *V. fischeri* to establish symbiosis with juvenile squid, *i.e.*, its symbiotic capacity. For pathogens, one common approach is to determine the infectious dose-50 ($ID_{50}$), which describes the amount of cultured pathogen that is necessary to infect half of the animals within a group. Because of the difficulty in precisely generating the inoculum that achieves this outcome, this metric is typically calculated by using multiple animal groups exposed to different concentrations of the infectious agent that together span the dynamic range of infection. In fact, previous studies of the *E. scolopes-V. fischeri* symbiosis have used this approach to evaluate different strains of *V. fischeri* [11, 18], but they used large numbers of animals.

A well-established method to circumvent the need for large sample sizes is that of Reed-Muench [19], which leverages the outcomes of animals across different inoculums to artificially inflate sample size in a manner that yields an $ID_{50}$. In this protocol, we apply the Reed-Muench method to quantify the symbiotic capacity of a *V. fischeri* strain through the calculation of the symbiotic dose-50 ($SD_{50}$), which corresponds to the inoculum concentration of *V. fischeri* that leads to symbiosis in half of a population of *E. scolopes* juveniles. Because experimentation with other animal-microbe symbioses also involves exposing the host to culture-grown cells [20, 21], we anticipate that the general approach described here to be broadly applicable to many other model systems.

This approach results in a $SD_{50}$ value using 42 juvenile squid (including the apo-symbiotic group), which is 2–5-fold fewer animals used to similarly assess a strain in previous reports [11, 18] (estimated as 90 and 240, respectively). In addition to lowering the number of animals used, this approach enables the assay to be performed more frequently using the animals produced by a mariculture facility. In a mariculture facility maintained in a 12-h light/12-h dark cycle, the juvenile squid hatch primarily during the light-to-dark transition [22], which determines the number of hatchlings available for experimentation on a given day. For instance, the daily juvenile squid production levels of an animal cohort were recently reported [13], with hatchlings produced over 80 days. Because there were 61 days for which at least 42 hatchlings were produced, the experiment described above could be performed 61 times. In contrast, the experiments involving either 90 or 240 animals could only be performed 48 times and 1 time,

respectively, because there were fewer days that the mariculture produced the appropriate number of hatchlings. Therefore, the approach described here also increases the utility of a given mariculture facility for evaluating the symbiotic capacity of *V. fischeri* strains, which enables researchers to perform more experiments and to determine whether the results are reproducible.

## Materials and methods

The protocol described in this peer-reviewed article is published on protocols.io (https://dx.doi.org/10.17504/protocols.io.yxmvm2155g3p/v2).

ES114 is a wild-type strain of *V. fischeri* [23]. Juvenile *E. scolopes* squid were generated by a mariculture facility as previously described [13].

Statistical analyses were determined either through Excel (see S1 and S2 Datasets) or Prism v. 9.5.1 (GraphPad Software, LLC). To identify the effect size corresponding to a specific power, the Goal Seek option of Excel was used.

## Expected results

Implementation of this protocol to determine the symbiotic capacity of a strain depends on using a range of inoculums over which the percentage of juvenile squid establishing symbiosis changes. Each group consists of seven hatchlings, which is one more than that initially reported by Reed and Muench [19], to account for a low but non-zero rate of animals that must be triaged due to poor health. For strains that have not been previously evaluated by this method, we suggest using 10-fold dilutions that range from $10^1$–$10^4$ CFU/mL to obtain an estimate for the $SD_{50}$ value. We also note that the inoculum duration used here (3.5 h) is commonly used by us and others to study ES114 because it is sufficiently long for cells to form the aggregates that promote host colonization [24, 25]. The duration does represent a variable that can be changed, but the resulting $SD_{50}$ values would likely be affected. Using a 3.5-h inoculum duration, we find that inoculums spanning 270–22,000 CFU/mL in 3-fold increments are sufficient to capture the dynamic range for assessing the symbiotic capacity of ES114 (Fig 1A).

This protocol requires each animal to first be scored for bioluminescence as the marker for symbiosis establishment. Bioluminescence arises in response to intracellular signaling that occurs both within and between the *V. fischeri* populations that assemble within the light organ [26, 27]. Consequently, the extent of bioluminescence emitted by an animal can vary based on cellular abundance, number of populations, and the ability of the strain to engage in intracellular signaling. However, the large effect size between symbiotic and non-symbiotic animals (Cohen's d = 8.85 for ES114, see S1 Dataset) enables us to generally use the 99.9[th] percentile of the apo-symbiotic control group as the cutoff for bioluminescence. These scores permit the classification of each animal as symbiotic or non-symbiotic, which are tabulated in a group according to each inoculum level (Table 1).

Application of the Reed & Muench method [19] results in columns of adjusted data that artificially inflates the sample size associated with each group (Fig 1B, 1C; Table 1), which enables a more precise estimation of the $SD_{50}$ for a strain without the need to increase the sample size of each group. We use the term adjusted symbiotic (non-symbiotic) animals in a manner analogous to the total dead (alive) nomenclature used by Reed & Muench [19]. For each inoculum, the corresponding adjusted symbiotic animal tally assumes that any animal that established symbiosis at lower inoculums would have also established symbiosis at the higher inoculum. Similarly, the adjusted non-symbiotic animal counts assume that animals that are non-symbiotic at high inoculums would also fail to establish symbiosis at lower inoculums.

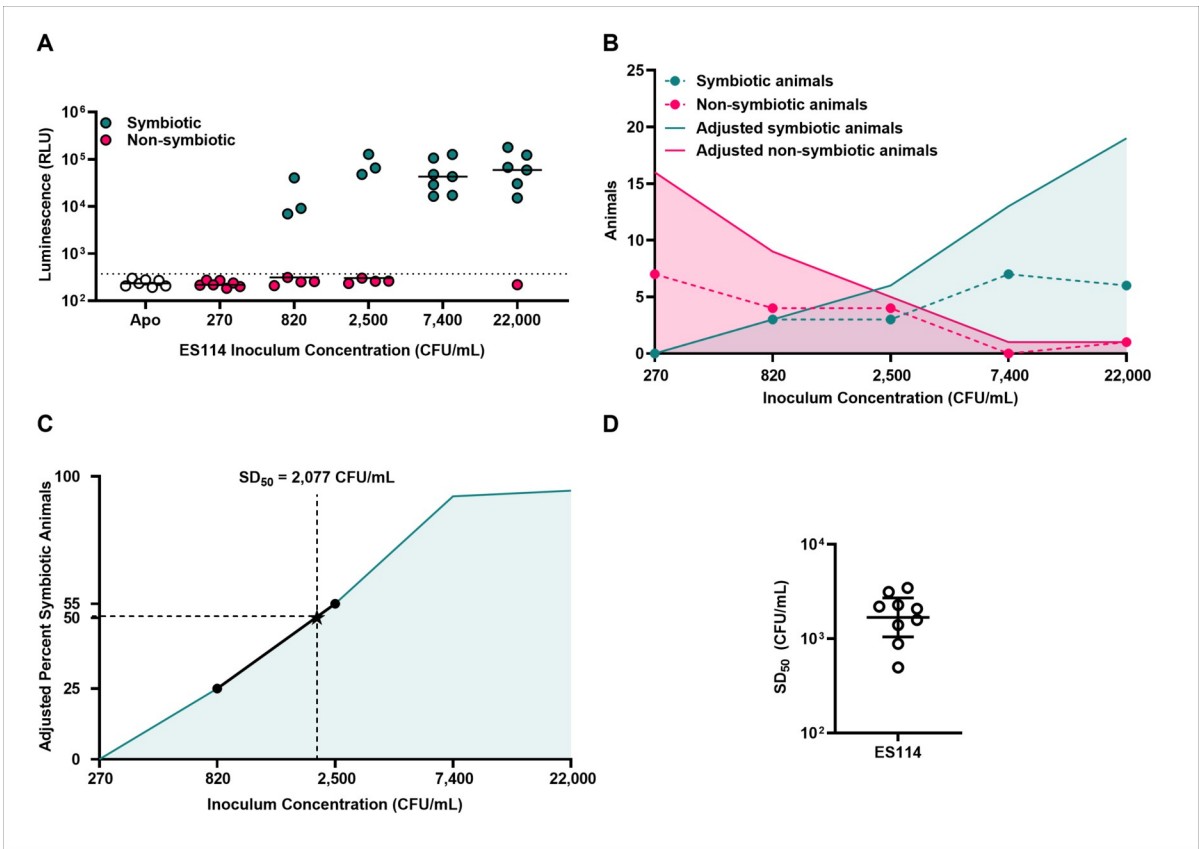

**Fig 1. Expected results. A)** Example squid-colonization assay showing the bioluminescence of animals at 20 h p.i. following a 3.5-h inoculation phase with the indicated amount of strain ES114. Each point represents an individual squid (N = 7 per group), and each bar represents the group median. Dotted line indicates the 99.9[th] percentile of the apo-symbiotic group (white symbols) used to score symbiotic (green) and non-symbiotic (magenta) squid. **B)** Graph of symbiotic and non-symbiotic animal counts listed in Table 1 form the experiment shown in A, with X-axis scaled to log base 3. Points indicate actual counts, and shaded areas highlight the counts following adjustments associated with the protocol. **C)** Graph of adjusted percent symbiotic animals from Table 1, with X-axis scaled to log base 3. Points indicate the inoculums used to calculate the $SD_{50}$, which is represented as a star. **D)** Values of $SD_{50}$ for ES114 from separate trials of the protocol. Each point represents an independent trial (N = 9 trials), and bars represent geometric mean ± 95% confidence intervals.

**Table 1. Counts of symbiotic and non-symbiotic animals shown in Fig 1A.**

| Inoculum Concentration (CFU/mL)[a] | Symbiotic Animals | Non-symbiotic Animals | Adjusted Symbiotic Animals[b] | Adjusted Non-symbiotic Animals[c] | Adjusted Percent Symbiotic Animals[d] |
|---|---|---|---|---|---|
| 22,000 | 6 | 1 | 19 | 1 | 95 |
| 7,400 | 7 | 0 | 13 | 1 | 93 |
| 2,500 | 3 | 4 | 6 | 5 | 55 |
| 820 | 3 | 4 | 3 | 9 | 25 |
| 270 | 0 | 7 | 0 | 16 | 0 |

[a]Dilution factor was 3.

[b]Sum of symbiotic animals from inoculum concentrations ≤ inoculum concentration of row.

[c]Sum of non-symbiotic animals from inoculum concentrations ≥ inoculum concentration of row.

[d]Percentage of symbiotic animals determined from adjusted values within row.

Using this approach, the $SD_{50}$ was determined to be 2,077 CFU/mL from the corresponding dataset using equations:

$$SD_{50} = 10^{(\log\ DF^x + \log\ c)} \tag{1}$$

$$x = \frac{50\% - a}{b - a} \tag{2}$$

where a = closest adjusted percent symbiotic below 50% (Table 1, 25%), b = closest adjusted percent symbiotic above 50% (Table 1, 55%), c = inoculum concentration of highest adjusted percent symbiotic below 50% (Table 1, 820 CFU/mL), and DF = dilution factor (3 for dataset shown in Table 1).

To test the approach for reproducibility, we performed eight additional experimental trials, which led to nine total calculations for the $SD_{50}$ of ES114 (see S2 Dataset). From these values, we conclude that the symbiotic capacity of ES114 is represented by an $SD_{50}$ of 1,684 CFU/mL with 95% confidence intervals of 1,047 and 2,709 CFU/mL (Fig 1D). This $SD_{50}$ is comparable to the ~1,250 CFU/mL $ID_{50}$ reported in one study [18] and approximately 6 times higher than another study that used natural seawater [11], which highlights that investigating how various components of seawater impact the $SD_{50}$ of a strain may identify environmental factors that affect symbiosis establishment. In practice, performing the experiment three times provides 80% power to detect a 2.7-fold change in the $SD_{50}$ (2-tailed Z-test, α = 0.05, see S2 Dataset).

## Supporting information

**S1 File. Step-by-step protocol.**
(PDF)

**S1 Dataset. Luminescence measurements of squid in experiment shown in Fig 1A.**
(XLSX)

**S2 Dataset. $SD_{50}$ values for ES114 shown in Fig 1D and corresponding power analysis.**
(XLSX)

## Acknowledgments

We thank Dr. Derek J. Fisher (Southern Illinois University) and members of the Miyashiro lab for helpful discussions regarding the protocol described here.

## Ethics declarations

Collection, care, and research of all laboratory animals was completed under the program's Institutional Animal Care and Use Committee (IACUC). IACUC protocol #PROTO202101789.

## Author Contributions

**Conceptualization:** Aidan R. Donnelly, Elizabeth J. Giacobe, Andrew G. Cecere, Tim I. Miyashiro.

**Data curation:** Aidan R. Donnelly, Elizabeth J. Giacobe, Rachel A. Cook, Gareth M. Francis, Grace K. Buddle, Christina L. Beaubrun.

**Formal analysis:** Aidan R. Donnelly, Elizabeth J. Giacobe, Tim I. Miyashiro.

**Funding acquisition:** Tim I. Miyashiro.

**Investigation:** Aidan R. Donnelly, Elizabeth J. Giacobe, Tim I. Miyashiro.

**Methodology:** Aidan R. Donnelly, Elizabeth J. Giacobe, Andrew G. Cecere, Tim I. Miyashiro.

**Project administration:** Tim I. Miyashiro.

**Resources:** Tim I. Miyashiro.

**Supervision:** Andrew G. Cecere, Tim I. Miyashiro.

**Validation:** Aidan R. Donnelly, Elizabeth J. Giacobe, Tim I. Miyashiro.

**Visualization:** Tim I. Miyashiro.

**Writing – original draft:** Aidan R. Donnelly, Elizabeth J. Giacobe, Tim I. Miyashiro.

**Writing – review & editing:** Aidan R. Donnelly, Elizabeth J. Giacobe, Andrew G. Cecere, Tim I. Miyashiro.

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
