## [Decision Letter · Decision Letter 0]

17 May 2023

PONE-D-23-08483Quantification of the Capacity of Vibrio fischeri to Establish Symbiosis with Euprymna scolopesPLOS ONE

Dear Dr. Miyashiro,

Thank you for submitting your manuscript to PLOS ONE. After careful consideration, we feel that it has merit but does not fully meet PLOS ONE’s publication criteria as it currently stands. Therefore, we invite you to submit a revised version of the manuscript that addresses the points raised during the review process. 

I am enclosing two positive and constructive reports by experts in the field, with which I overall agree. Below a brief summary of the major points raised by the reviewers (see further below for the full reviewer reports):

As it stands, the impact of this work is limited to the Squid-Vibrio system. Both reviewer have pointed out that the value of the here presented approach (specifically the experimental set-up and analytical part) for other colonization systems should be briefly discussed in the manuscript. This would also make the work more accessible for a broader audience.The general approach, data analysis and calculations (approach from Reed-Muench and final SD50 reported) sections would benefit from more detailed description.It would be helpful to include a clarification how the starting concentrations of Vibrio strains (in particular SD50 values) compare to what is available in the natural squid environment, and to what is typically used in other studies in the field.Finally, it would be helpful to clarify and discuss any potential correlation of squid bioluminescence with established symbiont titers (e.g., by potentially providing a calibration curve from sacrificed squid, if available/feasible).Please submit your revised manuscript by Jul 01 2023 11:59PM. If you will need more time than this to complete your revisions, please reply to this message or contact the journal office at plosone@plos.org. Please include the following items when submitting your revised manuscript:A rebuttal letter that responds to each point raised by the academic editor and reviewer(s). Please also explain cases where you disagree with the reviewers' or the editor's assessment. You should upload this letter as a separate file labeled 'Response to Reviewers'.A marked-up copy of your manuscript that highlights changes made to the original version. You should upload this as a separate file labeled 'Revised Manuscript with Track Changes'.An unmarked version of your revised paper without tracked changes. You should upload this as a separate file labeled 'Manuscript'.If applicable, we recommend that you deposit your laboratory protocols in protocols.io to enhance the reproducibility of your results. Protocols.io assigns your protocol its own identifier (DOI) so that it can be cited independently in the future. For instructions see: https://journals.plos.org/plosone/s/submission-guidelines#loc-laboratory-protocols. Additionally, PLOS ONE offers an option for publishing peer-reviewed Lab Protocol articles, which describe protocols hosted on protocols.io. Read more information on sharing protocols at https://plos.org/protocols?utm_medium=editorial-email&utm_source=authorletters&utm_campaign=protocols.

We look forward to receiving your revised manuscript.

Kind regards,

Claudia Isabella Pogoreutz

Academic Editor

PLOS ONE

Journal Requirements:

2. PLOS requires an ORCID iD for the corresponding author in Editorial Manager on papers submitted after December 6th, 2016. Please ensure that you have an ORCID iD and that it is validated in Editorial Manager. To do this, go to ‘Update my Information’ (in the upper left-hand corner of the main menu), and click on the Fetch/Validate link next to the ORCID field. This will take you to the ORCID site and allow you to create a new iD or authenticate a pre-existing iD in Editorial Manager. Please see the following video for instructions on linking an ORCID iD to your Editorial Manager account: https://www.youtube.com/watch?v=_xcclfuvtxQ.

Reviewers' comments:

Reviewer's Responses to Questions

**Comments to the Author**

1. Does the manuscript report a protocol which is of utility to the research community and adds value to the published literature?

Reviewer #1: Yes

Reviewer #2: Yes

2. Has the protocol been described in sufficient detail?

To answer this question, please click the link to protocols.io in the Materials and Methods section of the manuscript (if a link has been provided) or consult the step-by-step protocol in the Supporting Information files.

The step-by-step protocol should contain sufficient detail for another researcher to be able to reproduce all experiments and analyses.

Reviewer #1: Partly

Reviewer #2: No

3. Does the protocol describe a validated method?

Reviewer #1: Yes

Reviewer #2: Yes

4. If the manuscript contains new data, have the authors made this data fully available?

Reviewer #1: Yes

Reviewer #2: Yes

**5. Is the article presented in an intelligible fashion and written in standard English?**

Reviewer #1: Yes

Reviewer #2: Yes

6. Review Comments to the Author

Reviewer #1: Summary – relevance of the study

This manuscript focuses on the Squid-Vibrio symbiosis, a model system which has been key to study various aspects regulating the establishment of animal-bacteria associations, specifically those in which symbionts are acquired from the environment. The Lab Protocol describes a methodology to calculate the titers of bacteri, in this case a specific strain of Vibrio fischeri, required for colonization of Euprymna scolopes under laboratory conditions. There is a clear aim and justification for developing this procedure, which can be useful primarily for researchers working on the Euprymna – V. fischeri system, and potentially for those working on other animal-microbe symbioses. We detail a few points below that we believe could improve the manuscript.

General comments

- While the aim of the study is justified, the impact is rather limited to research groups using this model system. I believe that mentioning and discussing the value of such approaches for other symbiotic systems would improve the manuscript and make it more suitable for a broader audience, since the approach could be adapted accordingly.

- The data analysis and calculations could be explained more clearly (e.g. approach from Reed-Muench reference and final SD50 reported), please see the specific comments below.

- Is there evidence that the bioluminescence measured from a squid correlates directly with the established symbiont titers or a proof of principle established in other studies? If so, please provide reference(s). While in this paper a yes/no answer seems sufficient for the purpose, in my view this point is worth discussing and/or including in the manuscript, i.e. a calibration curve generated by plating suspensions from the sacrificed squids (dissected organs), of course, in case this is practically and ethically feasible with the euthanized individuals.

- It seems interesting and relevant to mention how the starting concentrations were determined and how these (specially the SD50) compare to the actual availability of this or other V. fischeri strains in the squid environment. I suggest mentioning this in the introduction (if previously reported) or elsewhere. While it is understandable that the results are not discussed in detail for this article format, it gives valuable context to the study for readers not working on the system.

Specific comments

- Lines 87-92: this was not entirely clear. Is this because of the total available juveniles per day/time-period, i.e. to carry out all replicates in a synchronized manner? Perhaps the authors can mention the reported number of squid production levels from ref 12 so it is easier to follow the rest of the paragraph? It is hard for non-squid experts to understand why the experiment can be performed 61/ 80 days and also in the following sentence. This could be elaborated on.

- Line 117: Reference for why 3.5 h inoculation phase was used?

- Lines 142-143: Although the reference is provided and the footnotes in Table 1 somewhat indicate the approach, I strongly suggest explaining this method more carefully and its validity in the main text as it is central to the study/proposed procedure.

- Line 156: How is this different from the 2,077 CFU/mL shown in Fig 1C? It also does not match the average SD50 across trials. This could be explained more specifically and/or adapted in the figure legend.

- Fig 1b: Could the lines be thicker? They are clearer in the TIF file but still hard to distinguish the green from pink, especially in the legend.

Supporting material:

SD50 protocol

1. Materials

- Item 7: do the authors mean 0.22 um filter? Please specify.

2. Selection and preparation of juvenile E. scolopes

- How are the animals transferred? Which tool(s) are needed?

3. Selection and Preparation of Juvenile E. scolopes

- For clarity could be rephrased as

"2) Prepare new tumblers with 50 ml FSSW for each group"

3) "Transfer animals from the 100 ml FSSW tumblers to the new 50 ml tumblers individually."

- Also, how many groups were there? This can be indicated here in relation to the number of different concentrations, for guidance.

9. Determination of Inoculum levels

- The data generated from this step should also be included in the expected results to get an impression of variation and deviations in the estimated inoculum numbers.

S1 Dataset: please indicate units

Reviewer #2: The manuscript from Donnelly, Giacobe, Miyashiro, and colleagues presents a method to efficiently calculate the 50% infectious dose during symbiotic colonization of bobtail squid by Vibrio fischeri. Although conceptually identical to the ID50, they term this the SD50 (symbiotic dose-50%) to note the application in this symbiotic context. The approach applies a method from Reed & Muench (1938) to try to use fewer animals than traditional approaches. The authors validate the method in nine independent replicates, which both shows the ease of applying the method and demonstrates the variation one can expect in applying the method.

Overall, this approach is exciting and represents a valuable contribution to the field. However, there are a few key items that are not described fully in the current version that would make it difficult for laboratories to adopt this approach. Additionally, I thought that some of the text introducing and framing the method was slightly misleading. Below are detailed comments.

Major points:

1. There seems to be insufficient description of the general approach. I am inferring from the tables and figures that one needs to use 7 squid per treatment and 3-fold dilutions of inoculum to gain the efficiency benefits of the approach used here, but neither detail is described in the manuscript or the protocol. Clear description and rationale for these numbers should be provided.

2. The description of the "Adjusted Symbiotic (or Non-Symbiotic) Animals" column in the data analysis is not clear and uses language distinct from the cited Reed & Muench (1938) paper. Only from the formulas embedded in the Excel sheet could I figure out how these numbers were generated. In addition to more clarity overall about what these numbers represent, it would also be helpful to connect these values to the language in the cited paper (e.g., We use the term Adjusted in the manner that Reed & Muench describe Total alive/dead).

3. How do the measured SD50 values from this method (Lines 156-157) compare to published results from other methods, and to inoculum concentrations that labs in the field use for colonization studies?

Minor point:

4. I think that the introduction is a bit misleading in that "Currently available protocols for this assay often lack justification for the abundance of V. fischeri cells within the inoculum" (Lines 25-26; similarly on 54-55). For example, the cited McCann et al. (2003) discusses this issue in detail, and other papers have cited that work in establishing their own inoculum concentrations. The current authors go on to discuss that evaluation of the ID50/SD50 value in different strains has not been established, and this seems like a more intellectually honest rationale for the work, especially as work on diverse natural V. fischeri natural isolates has increased (including work from the Miyashiro Lab).

5. I had a difficult time figuring out if this method was new to ID50 methods broadly or if it was only new to a beneficial symbiosis. I think that a clearer context for the work could better educate the readership. My current understanding is that this was originally described for ID50 work, and that the current study represents a novel application to beneficial symbioses. However, this approach seems to be rarely used in modern bacterial pathogenesis studies. Therefore, the authors may wish to note that while the focus of the work is on Vibrio fischeri-squid studies, the experimental setup and data analysis portions of the protocol could be applied broadly to other colonization systems.

7. PLOS authors have the option to publish the peer review history of their article (what does this mean?). If published, this will include your full peer review and any attached files.

Reviewer #1: **Yes: **Laura V. Florez

Reviewer #2: No

---

## [Author Response · Author response to Decision Letter 0]

20 May 2023

Please refer to the cover letter and response to reviewers file. Thank you.

---

## [Editor Report · Decision Letter 1]

7 Jun 2023

Quantification of the Capacity of Vibrio fischeri to Establish Symbiosis with Euprymna scolopes

PONE-D-23-08483R1

Dear Dr. Miyashiro,

We’re pleased to inform you that your manuscript has been judged scientifically suitable for publication and will be formally accepted for publication once it meets all outstanding technical requirements.

Kind regards,

Claudia Isabella Pogoreutz

Academic Editor

PLOS ONE
---

## [Editor Report · Acceptance letter]

4 Jul 2023

PONE-D-23-08483R1 

Quantification of the Capacity of *Vibrio fischeri* to Establish Symbiosis with *Euprymna scolopes*

Dear Dr. Miyashiro:

I'm pleased to inform you that your manuscript has been deemed suitable for publication in PLOS ONE. Congratulations! Your manuscript is now with our production department. 

Kind regards, 

on behalf of

Prof. Claudia Isabella Pogoreutz 

Academic Editor

PLOS ONE